# Comparative analysis of magnetically activated cell sorting and ultracentrifugation methods for exosome isolation

Eda Ciftci[1¤a], Naz Bozbeyoglu[2], Ihsan Gursel[2¤b], Feza Korkusuz[3], Feray Bakan Misirlioglu[4], Petek Korkusuz[5]*

1 Department of Bioengineering, Graduate School of Science and Technology, Hacettepe University, Ankara, Turkey, 2 Department of Molecular Biology and Genetics, Bilkent University, Ankara, Turkey, 3 Department of Sports Medicine, Faculty of Medicine, Hacettepe University, Ankara, Turkey, 4 Nanotechnology Research and Application Center, Sabancı University, Istanbul, Turkey, 5 Department of Histology and Embryology, Faculty of Medicine, Hacettepe University, Ankara, Turkey

¤a Current address: AO Foundation Research Institute Davos, Davos, Switzerland
¤b Current address: Izmir Biomedicine and Genome Center, Izmir, Turkey
* petek@hacettepe.edu.tr

**Data Availability Statement:** All relevant data are within the paper.

**Funding:** This study was funded by Hacettepe University Scientific Research Coordination Unit

## Abstract

Mesenchymal stem cell-derived exosomes regulate cell migration, proliferation, differentiation, and synthesis of the extracellular matrix, giving great potential for the treatment of different diseases. The ultracentrifugation method is the gold standard method for exosome isolation due to the simple protocol, and high yield, but presents low purity and requires specialized equipment. Amelioration of technical optimization is required for quick and reliable confinement of exosomes to translate them to the clinic as cell therapeutics In this study, we hypothesized that magnetically activated cell sorting may provide, an effective, reliable, and rapid tool for exosome isolation when compared to ultracentrifugation. We, therefore, aimed to compare the efficiency of magnetically activated cell sorting and ultracentrifugation for human mesenchymal stem cell-derived exosome isolation from culture media by protein quantification, surface biomarker, size, number, and morphological analysis. Magnetically activated cell sorting provided a higher purity and amount of exosomes that carry visible magnetic beads when compared to ultracentrifugation. The particle number of the magnetically activated cell sorting group was higher than the ultracentrifugation. In conclusion, magnetically activated cell sorting presents a quick, and reliable method to collect and present human mesenchymal stem cell exosomes to clinics at high purity for potential cellular therapeutic approaches. The novel isolation and purification method may be extended to different clinical protocols using different autogenic or allogeneic cell sources.

## Introduction

Nano-sized (30-150nm) extracellular vesicles (EVs), exosomes that are formed from multivesicular bodies contain proteins [1], sugars [2], nucleic acids [3], and lipids [4] of their cellular

[#TDK-2018-17530]. The funder had no role in study design, data collection, analysis, decision to publish, or preparation of the manuscript. The authors did not receive a salary from the founder.

**Competing interests:** The authors have declared that no competing interests exist.

source [5, 6]. The mesenchymal stem cells (MSCs) derived exosomes [7] which have great potential usage as a therapeutic agent instead of cell-based therapies, facilitate tissue repair via the regulation of cellular adhesion, proliferation, differentiation, extracellular matrix synthesis, and immune response [8, 9]. These kinds of exosomes are shown to be a good regenerative strategy for many pathological models such as wound healing [10, 11], peripheral nerve treatment [12, 13], liver healing [14], cardiac treatment [15], multiple sclerosis [16], and bone regeneration [17, 18].

The precipitation [19, 20], size exclusion chromatography [21, 22], ultrafiltration [23, 24], and ultracentrifugation [11, 25–33] present the principal techniques that are used for the isolation of exosomes [34]. The precipitation technique is an easy method [35, 36] but purity is a big challenge since non-exosome particles can precipitate within exosomes [37, 38]. The combination of precipitation and exclusion chromatography techniques [39, 40] is also another popular technique for isolation of exosomes. However, due to the particles of similar size coming from the participation method, it is not possible to say that there is a definitive solution for purity [41]. The efficiency of this technique depends on the manufacturer since reagents present high variability [42]. The size exclusion chromatography provides a purer exosome population for MSCs in a short time [43, 44] when compared to precipitation, but requires proceeding with other methods such as ultracentrifugation, which works with small volumes and intervenes with protein aggregation problems [45, 46]. Ultrafiltration is easy to handle, it has no volume limitation [43, 47, 48] and presents less protein contamination for exosomes [44] compared to precipitation [16, 49]. Nonetheless, the filter plugging [43, 50–52] and the exosome deformation or loss [53–55] problems could be encountered with the ultrafiltration technique. Ultracentrifugation is the most applied technique for exosome isolation that consists of serial centrifugation of samples at different speeds in which duration for separating the exosomes or extracellular vesicles from cells, cell debris, and apoptotic bodies are fixed [44, 56, 57]. This low-cost technique is simple to perform [44, 58] and allows to work with large volumes [44] but offers low purity compared to ultrafiltration [44, 59, 60]. The ultracentrifugation requires specialized and expensive equipment [43, 53].

Magnetically activated cell sorting (MACS) is an immunoaffinity-based technique that presents a new tool for exosome isolation from serum [61], plasma [62], urine [63], and several cancer cell lines [64–66] at high purity. This technique is based on the separation of exosomes from mixed solutions by their membrane-specific antibodies mainly the tetraspanins CD9, CD63, or CD81 [67, 68]. Recently, adipose tissue-derived MSCs-exosomes have been selected by magnetic beads using phosphatidylserine [68]. MACS may present a new instrument for exosome isolation since it enables the detection of selected subpopulations of exosomes and offers quite a pure population [64, 69]. However, it needs to be optimized to MSCs for maximum exosome yield and purity in order to increase their potential in clinical use. In this study, we asked whether tetraspanins specific MACS would present an optimized and effective technique for potential clinical application of exosomes when compared to commonly used ultracentrifugation. The aim of this study is to compare the efficiency of MACS by CD9, CD63 and CD81 antigens to ultracentrifugation for hMSCs-exosomes by using protein quantification, surface biomarker, particle size, particle number, and morphological analyses *in vitro*.

## Materials and methods

### Study design

A randomized, prospective *in vitro* study was designed in hMSCs-exosomes with 3 replicate 2 sets of experiments. The independent variables were determined as groups and dependent variables as (a) protein quantification, (b) particle size and number analysis by qNano, (c) purity

measurement, (e) surface marker analysis by flow cytometry and (d) morphological analysis by TEM. Exosomes that were isolated by MACS (n = 6) and ultracentrifugation (n = 6) constituted the experiment and control groups respectively.

## Cell culture

Human bone marrow-derived MSCs were procured (PCS-500-012, ATCC, USA) and preserved in DMEM supplemented with 10% FBS, 1% penicillin/streptomycin and incubated at 5% $CO_2$ at 37˚C. When cells reached 70% confluency, they were washed with PBS. After the washing process, DMEM supplemented with exosome depleted FBS (10%) was added to the flask, and cells were incubated at 5% $CO_2$ at 37˚C for 2 days. The manufacturer already reported and warranted the characterization of hBMSCs (Lot#63208778, ATCC, USA). During the whole experiment, MSCs were used in passage 4 as recommended by the manufacturer [70].

## Exosome isolation

**Ultracentrifugation.** Cell culture FBS exosome-depleted supernatant was filtered and transferred to sterile centrifuge tubes. Media was mixed with vortex and centrifuged at 1500xg for 10 minutes. The supernatant was transferred to an ultracentrifuge tube, completed with PBS, and centrifuged for 10 minutes at 10000xg by ultracentrifuge (XL-90 Ultracentrifuge, Beckman Coulter, USA) with SW 28 Ti Swinging-Bucket Aluminum Rotor (Beckman Coulter, USA). The supernatant was then centrifuged for 30 minutes at 30000xg. Pellet was removed and the supernatant was ultracentrifuged for 90 minutes at 100000xg. After completing the volume with PBS, the supernatant was aspirated, and the pellet was washed before being ultracentrifuged at 100000xg for 90 minutes. Finally, the supernatant was discarded, and the exosome-containing pellet was dissolved in 100 μL PBS.

**Magnetic activated cell sorting (MACS).** Exosome isolation with MACS Exosome Isolation Kit (#130-110-912, Miltenyi Biotec, Germany) was performed in obedience to the instruction of the manufacturer. Briefly, the cells, cell debris, and larger vesicles were removed from the cell culture media by centrifugation for 10 minutes at 300×g, 30 minutes at 2000×g, and 45 minutes at 10000×g consecutively. Then, 50 μl of CD9, CD63 and CD81 conjugated microbeads were added to 2 ml exosome-containing supernatant and vortexed. After mixing the beads and the sample, the mixture was held on incubation for 1 hour at room temperature. The μ-column was placed in the magnetic field of the μMACS separator and prepared by applying a 100 μl equilibration buffer on top of the column. Then, the column was rinsed with isolation buffer and the magnetically labeled sample was added onto the column and, the column was washed with isolation buffer. The column was removed from the magnetic separator and placed onto a 1.5 ml tube. After this step, 100 μl of isolation buffer was added to the column and the magnetically labeled vesicles were transferred to the 1.5 ml tube.

## Exosome characterization

**Determination of protein quantification.** Bicinchoninic acid (BCA) protein quantification (#23225, Thermo Fisher Scientific, USA) was performed [71]. Bovine serum albumin standards were prepared by serial dilutions (in between 2000 and 2 μg/ ml). Then, 25 μL serial standard solutions and samples were put into the 96-well plate and the working solution was added into the wells. After the incubation period for 30 minutes at 37˚C, absorbance was measured at 562 nm by a microplate reader (Synergy HT, Biotek, Winooski, VT, USA). This experiment was performed with 6 replicates.

**Exosome characterization by nanoparticle tracking analysis.** The particle numbers and particle size dispersity were identified by nanoparticle tracking analysis (NTA) (qNano Gold, Izon Science Ltd, New Zealand) [43]. Exosome samples isolated by ultracentrifuge and MACS were diluted to 1:20 and 1:2 respectively. NP80 nanopore which targets particles with 40-255nm diameter was used for analysis. The voltage was set to 0.56V and the particle counts began to be recorded when the current was stabilized. Two sets and 3 repeats of the experiment have been performed.

**Purity ratio measurement of exosome rich samples.** The ratio of particle number, obtained by NTA (n = 6), to protein concentration, determined by BCA (n = 6), presented the purity of exosome-rich samples [72–74]. Purity ratio calculated with the equation as follows;

$$\text{Purity ratio} = \frac{Particle\ concentration\ (P/ml)}{Protein\ concentration\ (\mu g/ml)} = \frac{Particle\ number\ (P)}{Protein\ amount\ (\mu g)}$$

**Exosome characterization by flow cytometry (FCM).** The carboxyl latex beads (#C37282, Thermo Fisher Scientific) were combined with antibodies of exosome surface markers [75]. The unconjugated anti-CD9 (#312102, Biolegend), and PE-labelled anti-CD81 (#349502, Biolegend) antibodies were used to capture exosomes. For the immune labeling process, 10 μL latex bead was completed to 500 μL with PBS and precipitated at 12000xg for 10 minutes. Pellet was dissolved in an anti-CD9 antibody so that the mixture was at 1μL bead for 1μg antibody, and the volume was completed to 100 μL with PBS. It was incubated while rotating at room temperature for 30 minutes. Incubated samples volume was completed to 500 μL with PBS and it was incubated overnight at room temperature. Bead and the antibody-containing solution was centrifuged at 12000xg for 10 minutes and dissolved in 1 ml 5% BSA solution. After incubation for 4 hours at room temperature for blocking, the mixture was centrifuged again at 12000xg for 10 minutes and the pellet was resuspended in 200 μL 1% BSA in PBS. Five micrograms of exosome were combined with 1 μL of the antibody-bead solution for staining. The final volume reached 50 μL with PBS and the mixture was held on 30 minutes incubation at room temperature. Upon incubation, the volume reached to 500 μL with PBS and the mixture was held on overnight incubation at room temperature while slowly rotating. The solution was precipitated at 12000xg for 10 minutes and the supernatant was discarded. Antibodies and their isotype control; mouse IgG1, κ PE (#400112, Biolegend, USA) were hold on 2 hours incubation with 1 μg/ml bead-exosome solution in 100 μL final volume at room temperature and avoid from the light. Incubated samples were washed with PBS. Samples were centrifuged at 12000xg for 10 minutes. The pellet was suspended in an appropriate volume of PBS and analyzed by flow cytometer (Novocyte, ACEA Biosciences, USA).

**Exosome characterization by transmission electron microscopy (TEM).** Isolated exosomes were diluted with PBS (1:100) pH:7.4 at room temperature and placed (3 μL) onto 200 mesh formvar/carbon-coated copper grids (EMS, USA). Samples were stained with 1% phosphotungstic acid and 2% uranyl acetate, dried at room temperature and observed under the TEM (Jeol, JEM 1400, Japan). The samples were quantitively evaluated at x50000 magnification and photographed by using a computer-attached digital camera (Gatan, USA) [74, 76].

## Statistical analysis

Shapiro Wilk test was used for evaluation of the normality of the distribution of the data obtained from analyses. Bonferroni and one-way ANOVA methods were used for the comparison of parametric data. Kruskal Wallis and Mann Whitney U tests for comparisons of

nonparametric outputs. The SPSS statistics software (v23, IBM, USA) was used for these analyses. The degree of significance was p <0.05.

## Results

### Protein quantification is determined by BCA analysis

The protein concentrations of MACS and ultracentrifuged isolated exosome sample were detected as 2.2 mg/ml and 1.8 mg/ml respectively when the results were normalized to 16 ml of starting sample. The particle number of the MACS group was higher when compared the that of the ultracentrifugation group, but the difference was not statistically significant (Fig 1A).

### Particle size and concentration is determined by NTA

The mean particle diameter of exosomes was recorded as 125 nm and 121 nm for MACS isolated and ultracentrifuged samples respectively. The maximum particle size range was found to be 88–426 nm with a peak of 110 nm in MACS isolated exosomes whereas it was noted as 80–383 nm with a peak of 110nm in ultracentrifuged exosomes. The mode of the particle diameter was detected as averagely 110 nm and 96 nm for MACS isolated and ultracentrifuged MSCs-exosomes respectively. The span value (d90-d10)/d50 reflecting the dispersity of particles was calculated as 0.52 and 0.60 for MACS isolated and ultracentrifuged particles respectively. The particle concentration of MACS isolated, and ultracentrifuged exosomes was 9.31 $\pm 4.4 \times 10^9$ particles/ml and $3.34 \pm 1.7 \times 10^9$ particle/ml respectively. The particle number of the MACS group was higher (p = 0.011) when compared to the ultracentrifugation (Fig 1B). All results were normalized to a 16 ml starting sample.

### Purity ratio of exosome-rich samples are measured

The ratio of purity measurement showed that MACS-isolated exosome-rich sample, $4.23 \times 10^9$ particle/mg was higher (p = 0.006) than that of ultracentrifuge $1.85 \times 10^9$ particle/mg (Fig 1C).

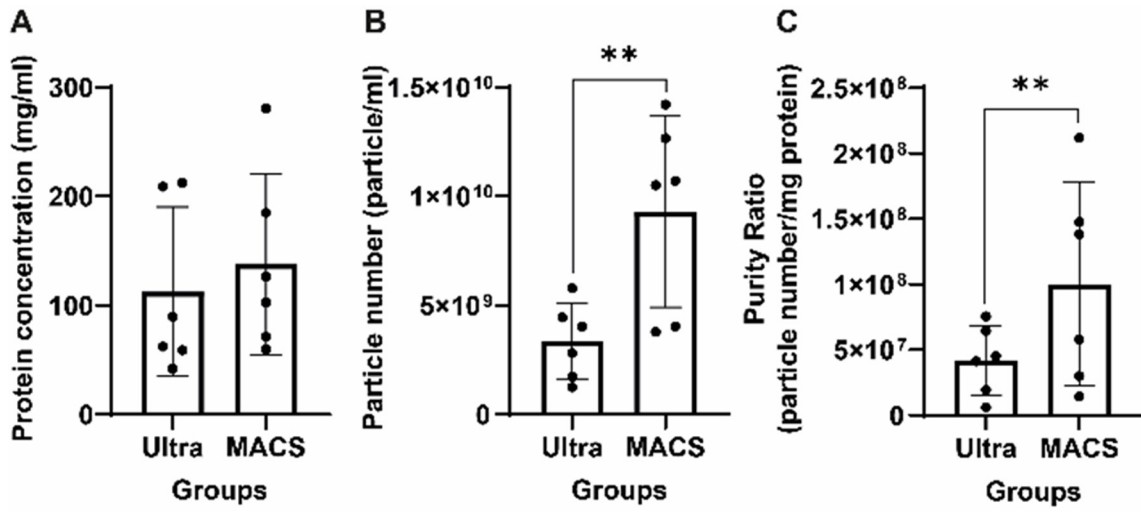

**Fig 1. Particle yield and purity analysis of exosomes.** Interleaved scatter bar graphic presents the descriptive data of (A) descriptive data of protein concentration (mg/ml) (B) particle number and (C) purity ratio analysis of MACS and ultracentrifugation groups. ** indicates p<0.05. (●) presented each individual measurement value.

### Exosome surface markers are analyzed by flow cytometry

MACS isolated hBMSCs-exosomes that captured by unconjugated-anti CD9 and labeled PE conjugated-anti CD81 antibodies exhibited 2-fold lower (p<0.001) mean fluorescence intensity (MFI) value (331) (Fig 2A) when comparing to ultracentrifuged -exosomes (632) (Fig 2B). Both ultracentrifuged and MACS isolated exosomes revealed higher (p<0.001) MFI values when compared to isotype control (Fig 2C).

### Morphological characterizations of isolated exosomes are occurred by TEM

Both the MACS and ultracentrifugation isolated exosomes exhibited negatively stained, spherical nano-sized particles. The samples that isolated MACS were in a range of sizes between 90–170 nm, and the ultracentrifuged samples were between 80–150 nm (Fig 3A–3C). The exosomes of both groups made round-shaped, intact particles or aggregated groups at the ultrastructural level. They have been distinct from the presence of the magnetic beads at MACS isolated samples (Fig 3B).

## Discussion

We isolated hMSCs-exosomes by MACS and ultracentrifugation and comparatively characterized by protein quantification, NTA, and TEM in this study. hMSCs-exosomes that isolated by MACS and ultracentrifuge, revealed similar protein quantity (2.2±0.17 mg/ml and 1.8±0.25 mg/ml, respectively), particle size (in a range between 80-170nm), and round-shaped morphology. The particle number of exosomes that were isolated by MACS (9.31±4.4x10$^9$ particles/ml) was higher than that of ultracentrifugation (3.34±1.7x10$^9$). MACS isolation technique provided 2.3-fold high purity to exosomes when compared to ultracentrifugation. MACS isolated exosomes exhibited a 2-fold lower MFI value that was 331 when compared to ultracentrifuged (632).

MACS and ultracentrifugation provided 2.2±0.17 mg/ml and 1.8±0.25 mg/ml protein quantity by BCA in this study. Previous studies on ultracentrifuged BMSCs-Exo samples revealed a protein quantity within a wide range between 0.27 mg/ml-1.18x10$^3$ mg/ml [14, 25, 77]. In a recent study, protein quantity was reported as 0.27 mg/ml in rat BMSCs-Exo samples

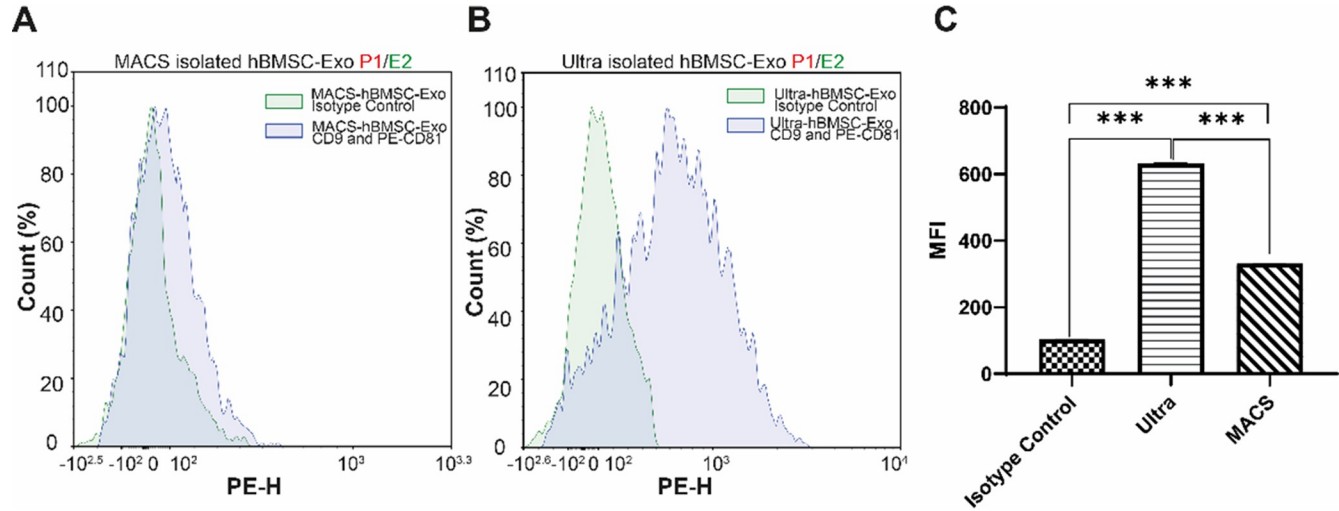

**Fig 2. Flow cytometric analysis of exosomes.** CD9 and CD81 surface expression on exosomes isolated by (A) MACS and (B) ultracentrifugation were analyzed by flow cytometry. (C) The MFI ratios of isotype control, ultracentrifuged, and MACS isolated- exosomes are shown. *** indicates p<0.001.

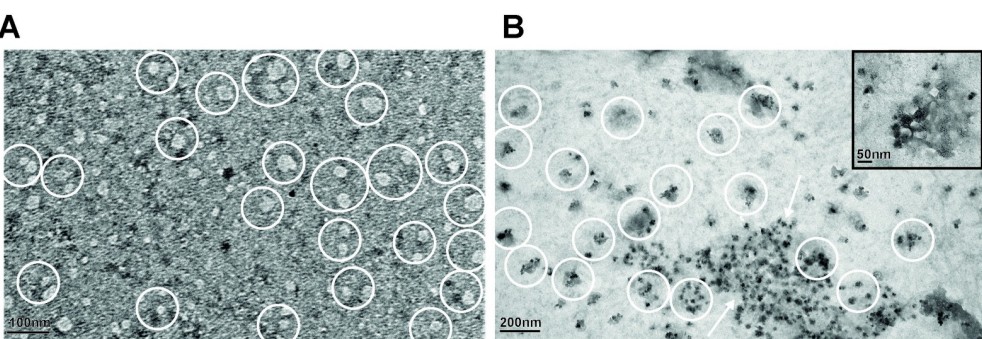

| Group | Size (nm) | Descriptive Statistics | | | | |
|---|---|---|---|---|---|---|
| | | Mean | Standard Dev. | Median | Min | Max |
| MACS | Width | 108.19 | 22.23 | 98.10 | 90.00 | 170.20 |
| | Length | 113.25 | 23.07 | 102.84 | 93.80 | 176.80 |
| Ultracentrifuge | Width | 105.98 | 20.87 | 102.3 | 80.12 | 150.21 |
| | Length | 111.14 | 19.31 | 106.8 | 88.24 | 151.23 |

**Fig 3. Transmission electron micrographs of exosomes.** Negatively stained spherical structures in a range of sizes between 80–150 nm and 90–170 nm at the (A) ultracentrifugation (scale bar:100nm) and (B) MACS (scale bar:200nm, inset 50nm) groups, respectively. Exosomes are marked with white circles and exosome clusters are pointed by arrows. Descriptive Statistics (C) of exosome sizes are shown.

[14]. Our protein quantity was about six to eight times higher than that study. Protein concentration was $1.18 \times 10^3$ mg/ml in another study that reported on ultracentrifuged hBMSCs-exosomes samples [77]. Our findings were in line with that study. Sajeesh et al reported hBMSCs-exosomes samples content as 1 mg/ml protein [25]. Different than our study, none of these studies reported their starting volume. This might be the reason of a wide range of protein quantity results in between these studies. BCA assessment should be supported by NTA analysis for particle number as it may not be solely related to the exosome purity [72, 73, 78].

MACS revealed a higher particle number when compared to ultracentrifugation by NTA in this study. Tan et al. [79] isolated hBMSCs-exosomes by phosphatidylserine labeled magnetic beads but did not compare their technique with ultracentrifugation. They reported the magnetically isolated hBMSCs-exosomes particle content as $4.1 \times 10^9$ particles per ml which contributes to the lower quantity when compared to our MACS samples ($9.31 \pm 4.4 \times 10^9$) [79]. Our exosome-specific triple surface marker selection (CD9, CD63, and CD81) on the other hand presented a more reliable method than the phospholipid capturing technique since it is based on the presence of three exosome specific tetraspanins [80, 81]. Apoptotic bodies with similar sizes with exosomes may be captured with magnetic beads when they are selected by phosphatidylserine which is a molecule that is shared by all extracellular vesicles [82]. Our ultracentrifuged samples revealed approximately 5-times higher particle numbers when compared to recent studies [25, 33, 83] reporting a range between 6 to $7.31 \times 10^8$ particle/ml and similar particle numbers when compared to Mead et al ($1.17 \times 10^9 \pm 1.42 \times 10^8$ particle per ml) [84] for hBMSCs-exosomes.

The ratio of protein to particle concentration determines the purity of exosome-rich samples [72, 73, 78]. Our results revealed the particle concentration to protein amount of exosomes as $4.23 \times 10^9$ by MACS and $1.85 \times 10^9$ by ultracentrifugation. The purity of human plasma exosomes was reported as $> 3 \times 10^{10}$ particle/mg protein by precipitation technique [72]. The human urinary exosomes exhibited a particle to protein ratio of $1 \times 10^9$ particle/mg by purity

ratio technique [73]. The particle to protein ratio of sucrose cushion-isolated prostate, urinary bladder and breast cancer cell lines [73] was reported as $5x10^9$-$4x10^{10}$ particle/mg by ultracentrifugation. There was no information on human BMSCs-Exo purity by MACS and ultracentrifuge methods in the literature. Weber et al stated the purity ratio range of cancer cell line exosome samples as highly pure when $> 3x10^{10}$ particle/mg protein, low purity when between $2x10^9$ and $2x10^{10}$ particle/mg and un-pure when less than $1.5x10^9$ particle/mg. In this study, purity of MACS isolated, and ultracentrifuged hBMSC-exosomes were categorized at low purity according to that classification. The source of exosomes may influence the ratio of purity. Healthy hBMSCs may release lower exosomes when compared to cancer cell lines and other body fluids. Nevertheless, MACS purity was higher than that of the ultracentrifugation as presented in this study.

In our study, exosomes isolated by MACS and ultracentrifuge had both low spans; 0.52 and 0.60 respectively. The span value provides the particle size distribution (PSD) and a smaller value means a homogenous sample with a similar particle size in the NTA system [72, 85]. Both samples were monodispersed whereas ultracentrifuge exosomes had more polydispersity when compared to MACS exosomes. Since MACS uses magnetic beads and lacks error-prone steps such as centrifugation, a little less polydispersity of MACS exosomes is understandable.

Here we additionally report the transmission electron microscopic appearance of exosomes that were isolated by MACS and ultracentrifugation as characteristic homogeneous, spherically shaped, and double membrane-layered structures. The MACS and ultracentrifugation isolated exosomes diameter range were noted in between 80 to 200 nm with a peak at 110 nm by NTA, and TEM revealed the size range between 90–170 nm, and 80–150 nm for MACS and ultracentrifugation respectively. Recent studies reported MACS isolated MSCs-exosomes having an average size in between 80.0±1.9 nm [68] and 170nm [79], with no information on protein quantity. Our particle size data by MACS and ultracentrifugation measured by TEM [32, 77, 86–90] and NTA [26, 32, 79, 83, 91] were in line with previous literature. The 80 to 200nm size range presents a correct and safe exosome size [26, 27, 32, 83, 92] that assures the safety of both methods.

MACS and flow cytometric (FCM) characterization and selection are based on the same triple tetraspanin markers CD 9, CD 63, and CD 81 that compete when binding. So, MACS and FCM characterization were not applicable together. This could limit the comparison of MACS performance with FCM characterization when applied to the same sample. Those tetraspanins are stated as general exosome markers and present the most reliable tool when used three together by MACS [67, 93] and FCM [80, 81].

Our data may have several limitations. The in vitro and in vivo efficiency tests might be made to compare the performance of MACS and ultracentrifugation techniques in the isolation and characterization of exosomes. This limitation, however, does not exclude future in vitro, in vivo, and clinical investigations because the statistical correctness of our study was confirmed. Furthermore, MACS and ultracentrifugation are reliable in vitro methods that should be evaluated for potential future customized therapies before the clinic. The study does not comprise any technical assessment of body fluids or other cell types since the objective is to isolate, characterize and expand autogenic or allogenic MSC exosomes from the bone marrow to use as potential cell therapeutics. Candidate exosomes from each cell type should be separately tested and optimized before clinical translation.

## Conclusion

The exosome isolation by MACS approach has many advantages such as providing isolation and characterization of exosomes on a single step, contributing high purity and rational yield;

and disadvantages such as low working volume and biomarker screening problems when compared to ultracentrifugation. Ultracentrifugation is regarded as a gold standard and expensive technique for isolation of exosomes allowing working with large volumes. In this study, we proposed MACS as a practical and reliable alternative isolation technique that improves purity and speed to ultracentrifugation for exosomes. MACS technique is a promising isolation method for potential clinical applications of autogenic or allogeneic mesenchymal stem cells in which time, reliability, repeatability, and practicality have vital importance.

## Author Contributions

**Conceptualization:** Eda Ciftci, Ihsan Gursel, Feza Korkusuz, Petek Korkusuz.

**Data curation:** Eda Ciftci.

**Formal analysis:** Eda Ciftci, Naz Bozbeyoglu.

**Funding acquisition:** Petek Korkusuz.

**Investigation:** Eda Ciftci.

**Methodology:** Eda Ciftci, Petek Korkusuz.

**Project administration:** Petek Korkusuz.

**Resources:** Petek Korkusuz.

**Supervision:** Feray Bakan Misirlioglu, Petek Korkusuz.

**Validation:** Eda Ciftci, Naz Bozbeyoglu.

**Visualization:** Eda Ciftci.

**Writing – original draft:** Eda Ciftci, Naz Bozbeyoglu.

**Writing – review & editing:** Ihsan Gursel, Feza Korkusuz, Feray Bakan Misirlioglu, Petek Korkusuz.

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
