## [Decision Letter · Decision Letter 0]

19 Dec 2022

PONE-D-22-28884Comparative analysis of magnetically activated cell sorting and ultracentrifugation methods for exosome isolation from human mesenchymal stem cell culture supernatantPLOS ONE

Dear Dr. Korkusuz,

Thank you for submitting your manuscript to PLOS ONE. After careful consideration, we feel that it has merit but does not fully meet PLOS ONE’s publication criteria as it currently stands. Therefore, we invite you to submit a revised version of the manuscript that addresses the points raised by the Reviewer.

We look forward to receiving your revised manuscript.

Kind regards,

Jacopo Sabbatinelli, MD, PhD

Academic Editor

PLOS ONE

Journal Requirements:

“The authors received no specific funding for this work.”

“Authors are grateful to Hacettepe University Research Fund for their support to this research [#TDK-2018-17530].”

 “The authors received no specific funding for this work.”

Reviewers' comments:

Reviewer's Responses to Questions

**Comments to the Author**

1. Is the manuscript technically sound, and do the data support the conclusions?

Reviewer #1: No

2. Has the statistical analysis been performed appropriately and rigorously? 

Reviewer #1: Yes

3. Have the authors made all data underlying the findings in their manuscript fully available?

Reviewer #1: Yes

4. Is the manuscript presented in an intelligible fashion and written in standard English?

Reviewer #1: Yes

5. Review Comments to the Author

Reviewer #1: Comparative analysis of magnetically activated cell sorting and ultracentrifugation methods for exosome isolation from human mesenchymal stem cell culture supernatant, an original work by Çiftci et al. reveal that Magnetically activated cell sorting provided a higher purity and amount of exosome when compared to ultracentrifugation. Magnetically activated cell sorting isolated samples exhibited the presence of magnetic beads morphologically. The particle number of the magnetically activated cell sorting group was higher than the ultracentrifugation. In conclusion, the isolation of exosomes derived from mesenchymal stem cells by magnetically activated cell sorting presents a quick and reliable method to collect them at high purity for clinical usage. The finding is interesting, however the abstract section should be improved. I provided some comments on the manuscript. Concerns should be addressed by authors.

1. English grammar and typo errors must to be corrected.

2. Rephrase and better the abstract.

3. Authors used a method to isolate exosomes, I think author should not specify type of exosomes. For example ‘’Optimal isolation does not exist yet for obtaining the exosomes derived from mesenchymal stem cells for clinical usage’’. So, exosomes isolation does not differ for different cells. Remove mesenchymal stem cells.

4. Authors did not use other cells, therefore they can not claim this method is better than Ultracentrifuge. Therefore rephrase conclusions and discussion.

5. Remove or transfer the paragraph from line 42 to 48. Valid and reliable isolation methods of exosomes need ………….

6. Rewrite keywords in short.

7. Authors use references in introduction (PMID: 33246476; PMID: 31979113; PMID: 34890589 ; PMID: 31291956 ; https://doi.org/10.1016/j.matdes.2020.109227)

8. Authors define Ultracentrifuge model and rotor used for exosomes isolation.

9. Did the author filter supernatants?

10. Uniform terms Mesenchymal Stem Cells or MSCs throughout the text.

11. Replace Fig. 3 with another better Figure.

12. Authors may use other body fluids (urine, plasma, serum,…) to compare these methods.

6. PLOS authors have the option to publish the peer review history of their article (what does this mean?). If published, this will include your full peer review and any attached files.

Reviewer #1: No

---

## [Author Response · Author response to Decision Letter 0]

1 Feb 2023

PONE-D-22-28884

Reviewer #1: Comparative analysis of magnetically activated cell sorting and ultracentrifugation methods for exosome isolation from human mesenchymal stem cell culture supernatant, an original work by Çiftci et al. reveal that Magnetically activated cell sorting provided a higher purity and amount of exosome when compared to ultracentrifugation. Magnetically activated cell sorting isolated samples exhibited the presence of magnetic beads morphologically. The particle number of the magnetically activated cell sorting group was higher than the ultracentrifugation. In conclusion, the isolation of exosomes derived from mesenchymal stem cells by magnetically activated cell sorting presents a quick and reliable method to collect them at high purity for clinical usage. The finding is interesting; however, the abstract section should be improved. I provided some comments on the manuscript. Concerns should be addressed by the authors. 

1. English grammar and typo errors must be corrected. 

We thank the reviewer for mentioning this point. English grammar and errors were checked and corrected. 

2. Rephrase and better the abstract. 

The abstract was revised accordingly as follows: 

LINES 25-47

Mesenchymal stem cell-derived exosomes regulate cell migration, proliferation, differentiation, and synthesis of the extracellular matrix, giving great potential for the treatment of different diseases. The ultracentrifugation method is the gold standard method for exosome isolation due to the simple protocol, and high but presents low purity and requires specialized equipment. Amelioration of technical optimization is required for quick and reliable confinement of exosomes to translate them to the clinic as cell therapeutics. In this study, we hypothesized that magnetically activated cell sorting may provide, an effective, reliable, and rapid tool for exosome isolation when compared to ultracentrifugation. We, therefore, aimed to compare the efficiency of magnetically activated cell sorting and ultracentrifugation for human mesenchymal stem cell-derived exosome isolation from culture media by protein quantification, surface biomarker, size, number, and morphological analysis. Magnetically activated cell sorting provided a higher purity and amount of exosomes that carry visible magnetic beads when compared to ultracentrifugation. The particle number of the magnetically activated cell sorting group was higher than the ultracentrifugation. In conclusion, magnetically activated cell sorting presents a quick, and reliable method to collect and present human mesenchymal stem cell exosomes to clinics at high purity for potential cellular therapeutic approaches. The novel isolation and purification method may be extended to different clinical protocols using different autogenic or allogeneic cell sources.

3. Authors used a method to isolate exosomes, I think the author should not specify the type of exosomes. For example, "Optimal isolation does not exist yet for obtaining the exosomes derived from mesenchymal stem cells for clinical usage". So, exosome isolation does not differ for different cells. Remove mesenchymal stem cells. 

We thank the reviewer for remarking on this unclear point. This concern was clarified and the abstract was revised accordingly as follows: 

LINES 30-35

Amelioration of technical optimization is required for quick and reliable confinement of exosomes to translate them to the clinic as cell therapeutics. In this study, we hypothesized that magnetically activated cell sorting may provide, an effective, reliable, and rapid tool for exosome isolation when compared to ultracentrifugation.

LINES 41-45

The particle number of the magnetically activated cell sorting group was higher than the ultracentrifugation. In conclusion, magnetically activated cell sorting presents a quick, and reliable method to collect and present human mesenchymal stem cell exosomes to clinics at high purity for potential cellular therapeutic approaches.

4. Authors did not use other cells, therefore they cannot claim this method is better than Ultracentrifuge. Therefore, rephrase conclusions and discussion. 

We thank the reviewer for the critical and important recommendation. Discussion and Conclusion parts of the manuscript have been revised accordingly by adding the limitation of the output in detail.

LINES 337-341

The study does not comprise any technical assessment of body fluids or other cell types since the objective is to isolate, characterize and expand autogenic or allogenic MSC exosomes from the bone marrow to use as potential cell therapeutics. Candidate exosomes from each cell type should be separately tested and optimized before clinical translation. 

LINES 350-353

MACS technique is a promising isolation method for potential clinical applications of autogenic or allogeneic mesenchymal stem cells in which time, reliability, repeatability, and practicality have vital importance. 

5. Remove or transfer the paragraph from lines 42 to 48. Valid and reliable isolation methods of exosomes need ………….

The paragraph between previous lines 42 to 48 new lines 48 to 54 was removed from the abstract according to the reviewer's recommendation. We thank the reviewer for mentioning this point. 

6. Rewrite keywords in short.

We thank the reviewer for mentioning this point. Keywords were reorganized and shortened according to the MESH index. 

7. Authors use references in the introduction (PMID: 33246476; PMID: 31979113; PMID: 34890589 ; PMID: 31291956 ; https://doi.org/10.1016/j.matdes.2020.109227) 

Recommended references were added to the introduction according to the reviewer's comment as following lines:

LINE 61

Nano-sized (30-150nm) extracellular vesicles (EVs), exosomes (Exos) that are formed from multivesicular bodies contain proteins [1], sugars [2], nucleic acids [3], and lipids [4] of their cellular source [5, 6] 

Reference 5: Rahbarghazi R, Jabbari N, Sani NA, Asghari R, Salimi L, Kalashani SA, et al. Tumor-derived extracellular vesicles: reliable tools for Cancer diagnosis and clinical applications. Cell Commun Signal. 2019;17(1):73.

Reference 6: Rezaie J, Ahmadi M, Ravanbakhsh R, Mojarad B, Mahbubfam S, Shaban SA, et al. Tumor-derived extracellular vesicles: The metastatic organotropism drivers. Life Sci. 2022;289:120216.

LINE 61

Especially, the mesenchymal stem cells (MSCs) derived exosomes [7] which have great potential use as a therapeutic agent instead...

Reference 7: Joo HS, Suh JH, Lee HJ, Bang ES, Lee JM. Current Knowledge and Future Perspectives on Mesenchymal Stem Cell-Derived Exosomes as a New Therapeutic Agent. Int J Mol Sci. 2020;21(3).

LINE 71

The precipitation (19, 20), size exclusion chromatography (21, 22), ultrafiltration (23, 24), and ultracentrifugation (11, 25-33) present the principal techniques that are used for the isolation of exosomes (34).

Reference 34: Nikfarjam S, Rezaie J, Zolbanin NM, Jafari R. Mesenchymal stem cell derived-exosomes: a modern approach in translational medicine. Journal of translational medicine. 2020;18(1):1-21.

LINE 73

Combination of precipitation and exclusion chromatography technique [39,40] is also another popular technique for the isolation of exosomes.

Reference 39: Feghhi M, Rezaie J, Akbari A, Jabbari N, Jafari H, Seidi F, et al. Effect of multi-functional polyhydroxylated polyhedral oligomeric silsesquioxane (POSS) nanoparticles on the angiogenesis and exosome biogenesis in human umbilical vein endothelial cells (HUVECs). Materials & Design. 2021;197:109227.

Reference 40: Zhao J, Xu L, Yang D, Tang H, Chen Y, Zhang X, et al. Exosome-driven liquid biopsy for breast cancer: recent advances in isolation, biomarker identification and detection. Extracellular Vesicle. 2022;1:100006.

We also add another citation in the introduction section between LINE 76:

However, due to the particles of similar size coming from the participation method, it is not possible to say that there is a definitive solution for purity [41] 

Reference 41: Böing AN, van der Pol E, Grootemaat AE, Coumans FA, Sturk A, Nieuwland R. Single-step isolation of extracellular vesicles by size-exclusion chromatography. J Extracell Vesicles. 2014;3.

8. Authors define the Ultracentrifuge model and rotor used for exosome isolation.

We thank the reviewer for addressing this missing point. The information was added to the related section as follows:

LINES 125-127

The supernatant was transferred to an ultracentrifuge tube, completed with PBS, and centrifuged for 10 minutes at 10000xg by ultracentrifuge (XL-90 Ultracentrifuge, Beckman Coulter, USA) with SW 28 Ti Swinging-Bucket Aluminum Rotor (Beckman Coulter, USA).

9. Did the author filter supernatants? 

We thank the reviewer for mentioning this missing point. Yes, we filtered the cell culture supernatants before centrifugation. The explanation was added to the related section as follows:

LINES 123-124

Cell culture FBS exosome-depleted supernatant was filtered and transferred to sterile centrifuge tubes.

10. Uniform terms Mesenchymal Stem Cells or MSCs throughout the text. 

We thank the reviewer for addressing this point. The abbreviations of mesenchymal stem cells were checked and uniformed as MSCs in the main text.

11. Replace Fig. 3 with another better Figure. 

Figure 3 was checked and the detailed inset section was added to Fig 3B to make beads clearly visible. We thank the reviewer for this precious comment. 

12. Authors may use other body fluids (urine, plasma, serum…) to compare these methods. 

We thank the reviewer for mentioning this important issue. We did not include the assessment of exosomes in body fluid since the main objective of the study is to use mesenchymal stem cells exosomes as the potential of cell therapeutics. The BMMSC exosomes are generally isolated and characterized are expanded from the BM of the patients and applied autogenously or from that of healthy donors and applied allogenously. We added this information as a brief sentence to the limitations paragraph of the discussion section. These changes were addressed as following lines: 

LINE 332-341

Our data may have several limitations. The in vitro and in vivo efficiency tests might be made to compare the performance of MACS and ultracentrifugation techniques in hBMMSC-exosome isolation and characterization. This limitation, however, does not exclude future in vitro, in vivo, and clinical investigations because the statistical correctness of our study was confirmed. Furthermore, MACS and ultracentrifugation are reliable in vitro methods that should be evaluated for potential future customized therapies before the clinic. The study does not comprise any technical assessment of body fluids since the objective is to isolate, characterize and expand autogenic or allogenic MSC exosomes from the bone marrow to use as potential cell therapeutics. Candidate exosomes from each cell type should be separately tested and optimized before clinical translation.

---

## [Decision Letter · Decision Letter 1]

10 Feb 2023

Comparative analysis of magnetically activated cell sorting and ultracentrifugation methods for exosome isolation

PONE-D-22-28884R1

Dear Dr. Korkusuz,

We’re pleased to inform you that your manuscript has been judged scientifically suitable for publication and will be formally accepted for publication once it meets all outstanding technical requirements.

Kind regards,

Jacopo Sabbatinelli, MD, PhD

Academic Editor

PLOS ONE

Additional Editor Comments (optional):

Reviewers' comments:

Reviewer's Responses to Questions

**Comments to the Author**

1. If the authors have adequately addressed your comments raised in a previous round of review and you feel that this manuscript is now acceptable for publication, you may indicate that here to bypass the “Comments to the Author” section, enter your conflict of interest statement in the “Confidential to Editor” section, and submit your "Accept" recommendation.

Reviewer #1: All comments have been addressed

2. Is the manuscript technically sound, and do the data support the conclusions?

Reviewer #1: Yes

3. Has the statistical analysis been performed appropriately and rigorously? 

Reviewer #1: I Don't Know

4. Have the authors made all data underlying the findings in their manuscript fully available?

Reviewer #1: Yes

5. Is the manuscript presented in an intelligible fashion and written in standard English?

Reviewer #1: Yes

6. Review Comments to the Author

Reviewer #1: Comparative analysis of magnetically activated cell sorting and ultracentrifugation

methods for exosome isolation an original work has been revised. Authors have addressed all concerns. Now it is acceptable.

7. PLOS authors have the option to publish the peer review history of their article (what does this mean?). If published, this will include your full peer review and any attached files.

Reviewer #1: No

---

## [Editor Report · Acceptance letter]

17 Feb 2023

PONE-D-22-28884R1 

Comparative analysis of magnetically activated cell sorting and ultracentrifugation methods for exosome isolation 

Dear Dr. Korkusuz:

I'm pleased to inform you that your manuscript has been deemed suitable for publication in PLOS ONE. Congratulations! Your manuscript is now with our production department. 

Kind regards, 

on behalf of

Dr. Jacopo Sabbatinelli 

Academic Editor

PLOS ONE